# Intratumoral Microbiota: Metabolic Influences and Biomarker Potential in Gastrointestinal Cancer

**DOI:** 10.3390/biom14080917

**Published:** 2024-07-27

**Authors:** Xueyuan Bi, Jihan Wang, Cuicui Liu

**Affiliations:** 1Department of Pharmacy, Honghui Hospital, Xi’an Jiaotong University, Xi’an 710054, China; 2Institute of Medical Research, Northwestern Polytechnical University, Xi’an 710072, China; jihanwang@nwpu.edu.cn; 3Department of Science and Education, Honghui Hospital, Xi’an Jiaotong University, Xi’an 710054, China

**Keywords:** intratumoral microbiota, gastrointestinal cancer, microbial biomarkers, tumor metabolism, tumor microenvironment

## Abstract

Gastrointestinal (GI) cancers impose a substantial global health burden, highlighting the necessity for deeper understanding of their intricate pathogenesis and treatment strategies. This review explores the interplay between intratumoral microbiota, tumor metabolism, and major types of GI cancers (including esophageal, gastric, liver, pancreatic, and colorectal cancers), summarizing recent studies and elucidating their clinical implications and future directions. Recent research revealed altered microbial signatures within GI tumors, impacting tumor progression, immune responses, and treatment outcomes. Dysbiosis-induced alterations in tumor metabolism, including glycolysis, fatty acid metabolism, and amino acid metabolism, play critical roles in cancer progression and therapeutic resistance. The integration of molecular mechanisms and potential biomarkers into this understanding further enhances the prognostic significance of intratumoral microbiota composition and therapeutic opportunities targeting microbiota-mediated tumor metabolism. Despite advancements, challenges remain in understanding the dynamic interactions within the tumor microenvironment (TME). Future research directions, including advanced omics technologies and prospective clinical studies, offer promising avenues for precision oncology and personalized treatment interventions in GI cancer. Overall, integrating microbiota-based approaches and molecular biomarkers into GI cancer management holds promise for improving patient outcomes and survival.

## 1. Introduction

Gastrointestinal (GI) cancers encompass a diverse group of malignancies affecting the digestive system, mainly including the esophagus, stomach, liver, pancreas, colon, and rectum (colorectum) [1,2,3]. These cancers collectively pose a significant global health burden, representing a leading cause of cancer-related morbidity and mortality worldwide. Each type of GI cancer presents distinct challenges in terms of diagnosis, treatment, and prognosis. For instance, colorectal cancer (CRC) is one of the most common GI malignancies, with risk factors including age, family history, dietary habits, and inflammatory bowel diseases (IBDs) [4]. On the other hand, pancreatic cancer (PCA) often presents at an advanced stage, leading to poor prognosis and limited treatment options [5]. Despite advancements in screening, early detection, and treatment modalities such as surgery, chemotherapy, radiation therapy, and targeted therapy, GI cancers remain associated with high mortality rates. Moreover, treatment resistance, disease recurrence, and metastasis pose formidable challenges in the management of these malignancies. Understanding the complex interplay between genetic, environmental, and lifestyle factors in the development and progression of GI cancers is critical for improving patient outcomes.

Intratumoral microbiota/microbiome refers to the diverse community of microorganisms residing within tumor tissues [6,7]. The composition of intratumoral microbiota is incredibly diverse, comprising bacteria, viruses, fungi, and archaea [8]. While the exact sources of these microorganisms are not fully understood, they may originate from the surrounding tissue, bloodstream, gut, or other organs. Additionally, some microbes may be attracted to the unique conditions within tumors, including hypoxia, altered pH levels, and nutrient availability [9]. Research into the role of intratumoral microbiota in cancer revealed both intriguing possibilities and complexities. Some studies suggest that certain microorganisms and their byproducts can promote inflammation, genomic instability, and immune suppression, all of which contribute to tumorigenesis [10,11]. However, there is also evidence to suggest that certain microbes may exert anti-tumor effects by stimulating immune responses against cancer cells [6].

Tumor metabolism, characterized by alterations in cellular energy production and nutrient utilization, plays a crucial role in cancer progression [12,13,14]. Cancer cells exhibit metabolic reprogramming to sustain their rapid proliferation, evade immune surveillance, and adapt to the tumor microenvironment (TME). One hallmark of tumor metabolism is aerobic glycolysis, also known as the Warburg effect, wherein cancer cells preferentially utilize glycolysis for energy production even in the presence of oxygen [15,16,17,18]. This metabolic shift allows cancer cells to generate biomass for proliferation and produce metabolites that support tumor growth and survival. In addition to glycolysis, cancer cells exhibit alterations in other metabolic pathways, including amino acid metabolism, lipid metabolism, and nucleotide biosynthesis [12,13,19]. These metabolic adaptations enable cancer cells to meet the increased demands for building blocks and energy required for sustained growth and proliferation. Furthermore, the TME influences tumor metabolism through factors such as nutrient availability, oxygen tension, and interactions with stromal cells and immune cells [20,21,22,23]. Dysregulated metabolism not only fuels tumor progression, but also contributes to therapeutic resistance and immune evasion. Understanding the metabolic dependencies of cancer cells and the TME is essential for developing targeted therapies that exploit metabolic vulnerabilities in cancer. Moreover, exploring the influence of intratumoral microbiota on tumor metabolism represents a novel avenue for therapeutic intervention in cancer treatment.

The gut microbiota (GM) emerged as a key player in GI health and disease, influencing various aspects of host physiology, including immune function, metabolism, and inflammation [24]. Dysbiosis, or imbalance in the GM composition, was implicated in the pathogenesis of GI disorders, including IBDs and GI cancers [25,26]. Given the close anatomical proximity between the gut and GI tumors, it is plausible that the intratumoral microbiota may influence the development and progression of GI cancers. Moreover, the unique metabolic characteristics of GI tumors, such as altered nutrient utilization and microenvironmental conditions, may create specific niches for microbial colonization and activity within the tumor. Investigating the influence of intratumoral microbiota on tumor metabolism in GI cancers holds significant promise for uncovering novel therapeutic targets and biomarkers. Understanding how microbial communities interact with tumor cells and modulate metabolic pathways may lead to the development of microbiota-based interventions to improve treatment outcomes and patient survival.

Overall, exploring the intricate interplay between intratumoral microbiota, tumor metabolism, and gastrointestinal cancer represents a promising area of research with the potential to advance our understanding of cancer biology and inform the development of personalized therapeutic strategies.

## 2. Intratumoral Microbiota and GI Cancer

### 2.1. Origin of Intratumoral Microbiota in GI Cancer

Recent studies identified three main sources of intratumoral microorganisms in GI cancers: mucosal barriers, adjacent normal tissues, and hematogenous spread [7,27]. Mucosal barrier disruption in cancers such as colorectal and pancreatic allows microorganisms from the gut and oral microbiota to invade tumors. Strong evidence suggests that intestinal microbes may be a significant source of intratumoral microbiota [28]. Cancers in the GI system have cavities exposed to the external environment, which favor microbial colonization. Adjacent normal tissues, once thought sterile, can harbor bacteria similar to those found in tumor tissues, aided by the immunosuppressive and hypoxic TME, which fosters microbial colonization. Hematogenous spread involves microorganisms from sites such as the mouth and intestines entering the bloodstream and reaching tumors via damaged blood vessels. For instance, in CRC, *Escherichia coli* (*E. coli*) can breach the gut vascular barrier, enter the bloodstream, and colonize the liver, promoting metastasis. Most tumor-associated bacteria and fungi are intracellular, often within cancer or immune cells, suggesting they may be transported as cellular fragments or intact cells, though direct infection through blood vessels cannot be excluded. The TME’s conditions support microbial growth and migration, allowing both “driver” bacteria such as *Bacteroides fragilis* and *Helicobacter pylori* (*H. pylori*), which facilitate carcinogenesis, and “passenger” opportunistic bacteria to thrive [29].

### 2.2. Recent Studies on the Potential Mechanisms and Biomarker Significance of Intratumoral Microbiota in GI Cancer

#### 2.2.1. Esophageal Cancer

The Cancer Microbiome Atlas (TCMA) provides a pan-cancer comparative analysis to distinguish tissue-resident microbiota from contaminants [30]. Recent studies leveraging TCMA data revealed distinct microbial signatures between upper (including esophageal and gastric cancer) and lower (including CRC) gastrointestinal tumors, identifying specific bacteria correlated with clinical characteristics of GI cancers [31]. In esophageal carcinoma (ESCA), TCMA analysis unveiled significant alterations in microbial composition between tumor and normal tissues, with notable changes observed in Firmicutes and Proteobacteria. Moreover, a microbial signature consisting of ten microbes was found to correlate with ESCA subtype, tumor stage, and survival status [32]. In esophageal squamous cell carcinoma (ESCC), investigations into the impact of intratumoral microbiota on neoadjuvant chemoimmunotherapy (NACI) response revealed that Streptococcus enrichment correlates with improved treatment outcomes, including increased CD8+ T-cell infiltration and prolonged disease-free survival. Notably, mouse experiments suggest that manipulating the microbiota could enhance the efficacy of immunotherapy. The presence of intratumoral microbiota is linked to the presurgical chemoimmunotherapy response in patients with ESCC. Specifically, the enrichment of Streptococcus signatures correlates with a favorable response to cancer immunotherapy. Additionally, fecal microbiota transplantation (FMT) may alter the composition of tumor-resident microbiota, enhancing the response to immunotherapy [33]. Intratumoral microbiome impacts immune infiltrates in TME and predicts prognosis in ESCC patients [34]. *Fusobacterium nucleatum* (*F. nucleatum*) functions as an oncogenic bacterium and can promote ESCC tumor progression. High *F. nucleatum* burden was associated with poor recurrence-free survival (RFS) in ESCC patients and correlated with reduced response to neoadjuvant chemotherapy, suggesting that targeting this bacterium may improve therapeutic outcomes in ESCC [35].

The microbiota of the distal esophagus are affected by acid reflux from the stomach. Acid reflux causes inflammation and mucosal damage, leading to alterations in the microbiome of the distal esophagus. This change allows columnar epithelium to replace the original squamous epithelium, potentially progressing to gastroesophageal reflux disease (GERD), Barrett’s esophagus (BE), and esophageal adenocarcinoma (EAC). In the upper part of the esophagus, the microbiota is influenced by oral resident flora, with *Porphyromonas gingivalis* (*P. gingivalis*) playing a role in promoting the development of ESCC [36,37].

#### 2.2.2. Gastric Cancer

*H. pylori* is a well-known carcinogen associated with gastric cancer (GC), documented extensively in epidemiological and clinical studies [38,39]. H. pylori infection in the stomach primarily contributes to GC through its ability to induce chronic gastritis and subsequent inflammatory cascades. The bacteria stimulate the production of pro-inflammatory cytokines such as IL-1β, TNF-α, and IL-6, creating a pro-tumorigenic environment [40]. Furthermore, chronic inflammation induced by microbial dysbiosis can result in the production of reactive oxygen species (ROS) and reactive nitrogen species (RNS). These reactive molecules can cause oxidative damage to DNA, proteins, and lipids, thereby contributing to mutagenesis and carcinogenesis. The persistent DNA damage and resulting genomic instability foster an environment conducive to malignant transformation and tumor development in the GI tract. Studies investigated the role of microbiome in GC development, recognizing *H. pylori*, while also identifying additional microbial species enriched in GC samples. Using a robust bioinformatics pipeline, the study found decreased microbial diversity in GC compared to nonmalignant tissue across two large cohorts. Moreover, distinct microbial enrichment patterns were observed among GC molecular subtypes [41]. Altered microbial diversity and enrichment of specific genera were observed in GC intratumoral microbiota, with Methylobacterium being significantly associated with poor prognosis and reduced CD8+ tissue-resident memory T (TRM) cells in the TME. The findings highlight the potential role of Methylobacterium in modulating TGFβ expression and CD8+ TRM cells in GC [42]. A recent study investigated the role of non-H. pylori gastric microbiota in stomach adenocarcinoma (STAD) by analyzing RNA sequencing, clinical, and DNA methylation data from The Cancer Genome Atlas (TCGA) project. The intratumoral microbiome profiles were associated with STAD occurrence, progression, and prognosis, with differential methylation changes observed in genes related to cancer pathways. Bi-directional mediation effects between intratumoral microorganisms and host DNA methylation were identified, highlighting their significance in cancer metastasis and prognosis. Additionally, cell experiments demonstrated that certain microorganisms, such as Staphylococcus saccharolyticus, could influence gastric cell proliferation and invasion. These findings underscore the intricate interplay between the intratumoral microbiome and host epigenetics in STAD progression and TME dynamics [43].

#### 2.2.3. Liver Cancer

As a microbe, hepatitis B virus (HBV) is one well-known risk factor of hepatocellular carcinoma (HCC) [44]. In recent years, the GM was recognized to significantly affect HCC development. However, the comprehensive characterization of the HCC tumor microbiome remained largely elusive. Li et al. employed metagenomic sequencing to characterize the intratumoral microbiota in HBV-related HCC [45]. More recently, Liu et al. found that HBV-related HCC exhibits distinct intratumoral microbiota and immune microenvironment signatures [46]. Using a combination of 16S rRNA fluorescence in situ hybridization (FISH), immunohistochemistry, and sequencing techniques, studies revealed distinct bacterial characteristics and metabolic profiles in HCC tissues compared to adjacent nontumor tissues [47]. Moreover, Huang et al. investigated the intratumoral microbiota associated with HCC progression, revealing increased diversity compared to normal liver tissue. Similarly, findings from another research also reveal higher microbial diversity in HCC tissues compared to adjacent tissues, with increased abundances of certain microorganisms such as Enterobacteriaceae and Fusobacterium, alongside decreased levels of antitumour bacteria such as Pseudomonas. Additionally, alterations in microbial metabolic pathways, particularly enhanced fatty acid and lipid synthesis, were implicated in influencing HCC progression [48]. Predominant phyla included Patescibacteria, Proteobacteria, Bacteroidota, Firmicutes, and Actinobacteriota, with specific taxa such as Streptococcaceae and Lactococcus marking HCC cirrhosis [49]. Additionally, studies demonstrated that the intratumoural microbiome signature can predict the prognosis of HCC [50,51].

Except for liver cancer, a study characterized the intratumor microbiome of intrahepatic cholangiocarcinoma (ICC) tissues. They identified a Gram-positive aerobic bacterium, Staphylococcus capitis, present within the tumors. Additionally, they found a higher abundance of Paraburkholderia fungorum in paracancerous tissues, which exhibited antitumor activity against ICC through modulation of alanine, aspartate, and glutamate metabolism, suggesting a potential therapeutic avenue for ICC treatment [52].

#### 2.2.4. Pancreatic Cancer

Pancreatic cancer (PCA) is an increasingly growing source of cancer-related deaths and is often diagnosed at advanced stages, and thereby resistant to conventional therapies [53]. The investigation of intratumoral microbiota in PCA represents a burgeoning field with profound implications for understanding tumorigenesis, disease progression, and therapeutic interventions [54,55]. Studies elucidated how microbiota and their products can influence the pancreatic TME, modulate the biological behavior of cancer cells, and impact immune system functionality. Intratumoral microbiota, including anaerobic bacteria such as Bacteroides, Lactobacillus, and Peptoniphilus, correlate with immune suppression and poor prognosis in pancreatic ductal adenocarcinoma (PDAC) by creating a hypoxic microenvironment that supports their growth and affects tumor metabolism [56]. While still in its infancy, research showed promising results in treating pancreatic cancer by targeting and modulating the intratumoral microbiota. Table 1 summarizes the recent studies on intratumoral microbiota in PCA research. These emerging evidences not only shed light on the complex interplay between microbiota and PCA, but also offer exciting prospects for developing innovative diagnostic and therapeutic strategies to combat this deadly disease.

#### 2.2.5. Colorectal Cancer

Colorectal cancer (CRC) is a multifactorial disease characterized by complex interactions between genetic, environmental, and microbial factors [73,74]. Emerging evidence suggests that the human GM plays a crucial role in CRC development and progression [75]. Specifically, the intratumoral microbiota garnered significant attention due to its potential influence on tumor biology, immune responses, and treatment outcomes. For instance, *F. nucleatum* is a dominant bacterial species in CRC tissue that is frequently associated with CRC and was implicated in promoting tumorigenesis through various mechanisms. *F. nucleatum* was reported as being associated with cancer progression and poorer patient prognosis of CRC [76]. Additionally, other bacterial species, such as *Bacteroides fragilis*, *E. coli*, and certain members of the Firmicutes phylum, were identified within CRC tumors [6], highlighting the diverse microbial composition within these tissues. Table 2 summarizes the recent studies on intratumoral microbiota in CRC research.

Intratumoral microbiota can contribute to CRC tumorigenesis and progression through various mechanisms. Firstly, certain bacterial species produce genotoxins, such as colibactin produced by some strains of *E. coli*, which can directly damage DNA. Colibactin induces double-strand breaks in the DNA, leading to mutations and genomic instability, which are critical steps in the initiation and progression of human CRC [77]. Additionally, chronic inflammation is a well-established risk factor for cancer, and intratumoral microbiota can significantly contribute to inflammatory responses within the GI tract. Persistent colonization by pathogenic bacteria can lead to continuous activation of the host immune response, resulting in chronic inflammation. In CRC, bacteria such as enterotoxigenic *Bacteroides fragilis* produce enterotoxins that trigger inflammation and promote tumorigenesis through the STAT3 signaling pathway [78]. The ongoing inflammation not only induces DNA damage and genomic instability, but also supports a microenvironment that promotes tumor growth and invasion. Moreover, intratumoral microbiota can influence CRC by activating carcinogenic pathways. For example, *F. nucleatum* can interact with cancer cells and modulate signaling pathways that promote tumorigenesis [79]. *F. nucleatum* adheres to epithelial cells through FadA adhesin, which activates β-catenin signaling, a pathway often implicated in CRC [80]. Activation of β-catenin signaling leads to increased cell proliferation, invasion, and resistance to apoptosis. Certain microbial metabolites, such as secondary bile acids produced by gut microbiota, can activate nuclear receptors such as farnesoid X receptor (FXR) and constitutive androstane receptor (CAR), influencing cell proliferation and survival pathways [81,82]. Furthermore, intratumoral microbiota can exert immunosuppressive effects that facilitate tumor growth and metastasis in CRC. *F. nucleatum* promote the recruitment of myeloid-derived suppressor cells (MDSCs) and regulatory T cells (Tregs), which suppress effective immune responses against the tumor [83]. This immunosuppression hampers the host’s capacity to recognize and attack tumor cells, thereby facilitating cancer progression.

**Table 2 biomolecules-14-00917-t002:** Summary of recent studies on intratumoral microbiota in colorectal cancer.

Year of Study	Nature of Study	Participants	Aim and Main Findings	Refs.
2024	Observational study	CRC patients	This study aimed to elucidate the genetic factors of *F. nucleatum* facilitating tumor colonization in CRC by analyzing closed genomes of 135 *F. nucleatum* strains. It identifies a distinct clade, Fna C2, as predominant in CRC tumors, showing increased metabolic potential and colonization of the GI tract, and providing insights into the pathoadaptation of Fna C2 to the CRC tumor niche.	[76]
2023	Observational study	CRC patients	The study proposes a method to detect bacterial signals in human RNA sequencing data and associates them with clinical and molecular properties of tumors. The analysis reveals correlations between intratumoral microbiome composition and survival, anatomic location, microsatellite instability, consensus molecular subtype, and immune cell infiltration in colon tumors.	[84]
2024	Prospective-Observational study	CRC patients	This study investigates the tumor microbial profile of young-onset CRC (yoCRC) compared to average-onset CRC (aoCRC), revealing higher microbial diversity and distinct microbial compositions in yoCRC tumors. Akkermansia and Bacteroides are enriched in yoCRC tumors, while aoCRC tumors show more abundances of several other bacteria.	[85]
2023	Observational study	Patients with locally advanced rectal cancer	This study investigates the tumor-bearing microbiota in patients with locally advanced rectal cancer before neoadjuvant chemoradiation therapy (nCRT) and its association with treatment response. The findings reveal specific microbial biomarkers and functional pathways associated with resistance to nCRT, highlighting the potential role of intratumoral microbiota in modulating treatment outcomes in rectal cancer patients.	[86]
2023	Observational study	CRC patients (data from the TCGA)	The aim of this study was to unravel the potential remodeling mechanisms of immune cell infiltration and tumorigenesis in CRC by integrating genetic, epigenetic, and intratumor microbial factors. Results reveal the significant influence of intratumor microbes on immune cell infiltration patterns, prognosis, and response to immune checkpoint blockade therapy in CRC.	[87]
2023	Observational study	Locally advanced rectal cancer (LARC) patients (data from a published European cohort)	This study investigates the intratumoral microbiota in LARC patients and its association with the response to nCRT. It identifies microbial signatures associated with pathological complete response (pCR) and non-pCR groups, highlighting their potential as independent predictive markers for nCRT response and revealing interactions between intratumoral microbes and cancer-associated fibroblasts (CAFs) in mediating treatment response.	[88]
2024	Observational study	CRC patients	The study aimed to understand how Colibactin-producing *E. coli* influences tumor heterogeneity, chemoresistance, and patient survival in right-sided CRC tumors. The main findings show that Colibactin-producing *E. coli*-infected tumors had high glycerophospholipid environments, reduced CD8+ T lymphocyte infiltration, and increased chemoresistance through lipid droplet accumulation and phosphatidylcholine remodeling.	[89]
2021	Observational study	CRC patients	The study explored the association between the intratumor microbiome and host genetic alterations in CRC patients. Fusobacterium was associated with mutated genes and cell cycle-related pathways, while Campylobacter abundance was linked to mutational signature 3, suggesting a potential role of bacterial-induced DNA damage in CRC.	[90]
2017	Observational study	Microsatellite instability-high (MSI-H) CRC patients	The study aimed to investigate the clinicopathologic and molecular associations of *F. nucleatum* in MSI-H CRC patients. High intratumoral *F. nucleatum* were associated with increased macrophage infiltration and CDKN2A promoter methylation in MSI-H CRC.	[91]
2021	Observational study	MSI-H CRC patients	In MSI-H CRC, high levels of intratumoral *F. nucleatum* are associated with larger tumor size and advanced invasion depth. Additionally, *F. nucleatum*-enriched tumors exhibit decreased density of FoxP3+ T cells and an increased proportion of M2-polarized macrophages in the tumor center.	[92]
2018	Observational study	CRC patients	The aim of this study was to investigate the association between the amount of Bifidobacteria in CRC tissue and tumor differentiation, specifically the extent of signet ring cells, as well as the immune response to CRC. The main findings reveal that intratumor bifidobacteria were detected in 30% of cases and were associated with the extent of signet ring cells, suggesting a possible role of bifidobacteria in determining distinct tumor characteristics or as an indicator of dysfunctional mucosal barrier in CRC.	[93]
2019	Observational study	CRC patients	The aim of this study was to investigate the prognostic impact of intratumoral *F. nucleatum* in CRC patients treated with adjuvant chemotherapy. Intratumoral *F. nucleatum* load was found to be a potential prognostic factor in stage II/III CRC patients treated with oxaliplatin-based adjuvant chemotherapy, particularly in non-MSI-H/non-sigmoid/non-rectal cancer subsets.	[94]

## 3. Role of Intratumoral Microbiota in Modulating GI Tumor Metabolism

### 3.1. Influence of Microbial Dysbiosis on Tumor Metabolic Reprogramming

Dysbiosis, or the disruption of microbial balance, can lead to alterations in microbial composition and function, influencing tumor metabolism [95]. Dysbiotic conditions within the TME can trigger metabolic reprogramming in cancer cells, promoting tumor growth, invasion, and metastasis [96,97]. This dysregulated metabolism often involves shifts in key metabolic pathways, including glycolysis, fatty acid metabolism, and amino acid metabolism, which support the energetic and biosynthetic demands of rapidly proliferating cancer cells.

### 3.2. Specific Metabolic Pathways Affected by Intratumoral Microbiota in GI Cancer

#### 3.2.1. Glycolysis

Recently, studies reported the crosstalk between intratumoral bacteria and the tumor. In the immediate TME, marked by vascular hyperplasia, aerobic glycolysis, hypoxia, and immunosuppression, bacterial proliferation becomes favorable. Intratumoral bacteria, integral to this milieu, significantly impacts tumor progression, metastasis, and the efficacy of anti-tumor treatments [98]. The intricate interplay between the intratumoral microbiota and tumor metabolism in GI cancer involves modulation of specific metabolic pathways critical for cancer progression and therapeutic response. One such pathway is glycolysis, the process by which glucose is metabolized to produce energy in the form of ATP. Dysbiosis within the TME can promote glycolytic flux in cancer cells, leading to increased glucose uptake and lactate production, a phenomenon known as the Warburg effect. For instance, the normal esophageal microbiome, which is primarily composed of Gram-positive bacteria, undergoes a shift to a predominantly Gram-negative microbiome during the dysbiosis associated with the EAC cascade. This altered microbiota can then promote the pathogenesis of EAC by activating toll-like receptors (TLRs), inducing cyclooxygenase-2 expression, stimulating inducible nitric oxide synthase, activating the NLRP3 inflammasome, and contributing to the Warburg effect [37]. Additionally, PDAC is known to create a hypoxic microenvironment due to its limited cellularity and dense, desmoplastic stroma [99,100]. Hypoxia in PDAC was shown to enhance the intracellular survival of anaerobic bacteria such as *P. gingivalis* [101]. Recent research also indicates that intestinal bacteria can intensify the hypoxic conditions, which in turn modulates tissue-resident lymphocytes [102]. Consequently, the tumor stroma and the intratumoral colonization of bacteria may establish the hypoxic environment that supports the growth of anaerobic bacteria in PDAC cases, potentially leading to poor prognosis through immune suppression. Certain bacterial species, such as *F. nucleatum*, were implicated in promoting glycolysis in CRC cells by upregulating glucose transporters and glycolytic enzymes [103,104]. This metabolic shift towards glycolysis not only provides cancer cells with a sustained energy supply, but also contributes to the acidic TME, fostering tumor growth and immune evasion.

#### 3.2.2. Fatty Acid Metabolism

Another metabolic pathway influenced by intratumoral microbiota is fatty acid metabolism, which plays a crucial role in providing cancer cells with essential lipids for membrane synthesis, energy storage, and signaling [47,105]. Dysbiosis-associated alterations in lipid metabolism were observed in GI tumors, with certain bacterial species implicated in promoting lipogenesis and lipid accumulation within cancer cells [106]. For example, *F. nucleatum* was shown to induce expression of fatty acid synthase (FASN), a key enzyme involved in de novo lipogenesis in CRC cells [107]. This dysregulated lipid metabolism not only fuels cancer cell proliferation and survival, but also contributes to tumor progression and chemoresistance.

#### 3.2.3. Amino Acid Metabolism

Furthermore, amino acid metabolism represents another metabolic pathway influenced by the intratumoral microbiota in cancer pathogenesis and immunity [27]. Amino acids serve as crucial building blocks for protein synthesis and also play diverse roles in cellular metabolism and signaling pathways. Dysbiosis-induced alterations in amino acid metabolism were linked to tumor progression and therapeutic resistance in GI cancers. For instance, certain bacterial species can modulate the availability of specific amino acids, such as glutamine and arginine, within the TME, thereby impacting cancer cell metabolism and immune cell function [108]. Additionally, dysbiosis-associated alterations in amino acid metabolism can contribute to the generation of immunosuppressive metabolites, such as kynurenine, which suppress anti-tumor immune responses and promote immune evasion in tumors [109,110,111].

Overall, the relationship between intratumoral microbiota and tumor metabolism in GI cancers is complex and multifaceted. Intratumoral microbiota can influence tumor metabolism by producing metabolites such as short-chain fatty acids (SCFAs), which alter gene expression and metabolic pathways in cancer cells. In GI cancers, tumor cells often undergo metabolic reprogramming, increasing glycolysis, glutaminolysis, and lipid metabolism. This metabolic environment promotes the growth of specific microbiota, which in turn produce metabolites that further support tumor metabolism and proliferation. Moreover, intratumoral microbiota can affect the TME, influencing the efficacy of metabolic therapies and immunotherapies. Figure 1 summarizes the complex interactions between intratumoral microbiota and tumor metabolism in GI cancers. Additionally, different types of GI cancers exhibit unique patterns of microbiota metabolism interactions, which require further study to understand fully.

## 4. Clinical Implications of Intratumoral Microbiota-Mediated Tumor Metabolism in GI Cancer

### 4.1. Prognostic Significance of Intratumoral Microbiota Composition

The composition of intratumoral microbiota emerged as a potential prognostic marker in GI cancer, offering valuable insights into disease progression and patient outcomes [11,112]. As described above, numerous studies reported associations between specific microbial species or dysbiotic patterns and clinical outcomes in GI cancer patients. For instance, higher abundance of *F. nucleatum* is consistently associated with poorer prognosis in colorectal CRC, including increased risk of recurrence, metastasis, and reduced overall survival. Similarly, alterations in the composition of intratumoral microbiota, such as decreased microbial diversity or enrichment of pathogenic bacteria, were linked to adverse clinical outcomes in various GI malignancies. Understanding the prognostic significance of intratumoral microbiota composition holds promise for improving risk stratification and informing personalized treatment strategies in GI cancer patients.

### 4.2. Therapeutic Opportunities Targeting Microbiota-Mediated Tumor Metabolism in GI Cancer

The influence of intratumoral microbiota on tumor metabolism presents novel therapeutic opportunities for the treatment of GI cancer. Targeting microbiota-mediated tumor metabolism represents a promising approach to disrupt cancer progression and enhance therapeutic efficacy in GI malignancies. Several strategies were proposed to modulate intratumoral microbiota and tumor metabolism for therapeutic benefit. These include the use of probiotics, prebiotics, antibiotics, and microbial-based therapies to manipulate the composition and function of intratumoral microbiota. Additionally, targeting specific metabolic pathways dysregulated by microbiota within the TME holds potential for developing innovative treatment modalities. For example, inhibitors targeting key enzymes involved in microbial-induced metabolic reprogramming, such as FASN or glycolytic enzymes, may offer therapeutic benefits in GI cancer. Furthermore, strategies aimed at restoring metabolic homeostasis and immune function within the TME, such as metabolic inhibitors or immunotherapies, hold promise for overcoming microbiota-mediated immune evasion and enhancing antitumor immune responses in GI cancer. To further illustrate these therapeutic and diagnostic methodologies, a schematic pipeline from research to clinical application is provided in Figure 2. This pipeline demonstrates how intratumoral microbiota can be leveraged for therapeutic and diagnostic purposes in GI cancer. Overall, therapeutic targeting of microbiota-mediated tumor metabolism represents a promising avenue for the development of novel treatment strategies in GI cancer, with the potential to improve patient outcomes and survival.

## 5. Challenges and Future Directions

### 5.1. Current Limitations in Understanding the Complex Interplay between Intratumoral Microbiota and Tumor Metabolism

Despite significant advancements, several challenges hinder our comprehensive understanding of the intricate relationship between intratumoral microbiota and tumor metabolism in GI cancer. One major limitation is the complexity of the TME, which comprises diverse microbial communities interacting with tumor, stromal, and immune cells [113,114]. Characterizing the spatial distribution and functional activities of intratumoral microbiota within this complex ecosystem remains challenging, requiring advanced techniques and multidisciplinary approaches. Additionally, the dynamic nature of microbial communities and tumor metabolism poses challenges in deciphering causal relationships and temporal dynamics [115,116]. Furthermore, the heterogeneity of GI cancers, both within and between patients, adds another layer of complexity to studying microbiota-mediated tumor metabolism.

### 5.2. Future Research Directions for Unraveling the Mechanisms and Clinical Applications in GI Cancer

Several promising research directions hold potential for advancing our understanding of intratumoral microbiota-mediated tumor metabolism and translating these insights into clinical applications for GI cancer. Firstly, applying advanced omics technologies, such as metagenomics, metabolomics, and single-cell sequencing, will enable comprehensive profiling of intratumoral microbiota and tumor metabolism at high resolution [117]. Integrative analyses of multi-omics data will provide insights into the functional interactions between microbiota, tumor cells, and the TME. Moreover, mechanistic studies focusing on specific microbial metabolites and metabolic pathways implicated in GI cancer progression will elucidate the underlying molecular mechanisms and identify potential therapeutic targets. Additionally, prospective clinical studies with larger cohorts and longitudinal follow-up are needed to validate the prognostic and predictive value of intratumoral microbiota composition and metabolic signatures in GI cancer. Finally, the development of microbiota-targeted therapeutics and precision medicine approaches tailored to individual patient profiles will improve treatment outcomes and patient survival in GI cancer. Overall, addressing these research priorities will be beneficial for harnessing the potential of intratumoral microbiota in guiding precision oncology strategies and personalized treatment interventions in GI cancer.

## 6. Conclusions

In conclusion, this review highlighted the significant impact of intratumoral microbiota on tumor metabolism and its potential as a biomarker in five major types of GI cancer, as illustrated in Figure 1. Through the summary of current research, we identified key findings that demonstrate how microbial dysbiosis within the tumor microenvironment (TME) can affect cancer progression, molecular mechanisms, and treatment outcomes. Furthermore, we emphasize the importance of integrating microbiota-based approaches and molecular biomarkers into GI cancer management strategies. By understanding the complex interplay between intratumoral microbiota, tumor metabolism, and molecular mechanisms, clinicians and researchers can develop novel therapeutic interventions and precision medicine strategies tailored to individual patients. Overall, this comprehensive understanding opens new avenues for targeted therapies and personalized treatment modalities, ultimately improving outcomes for patients with GI cancer.

## Figures and Tables

**Figure 1 biomolecules-14-00917-f001:**
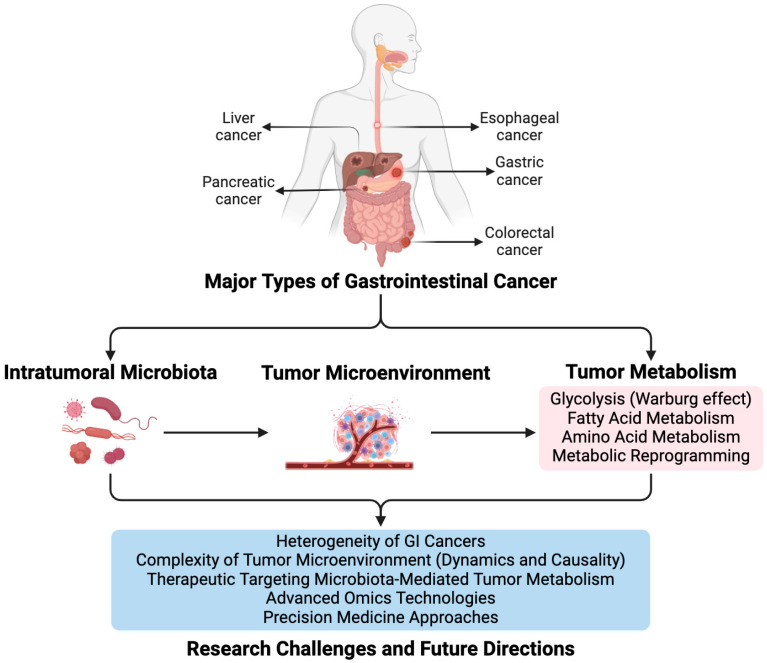
Summary of Intratumoral Microbiota’s Influence on Tumor Metabolism in Major Types of GI Cancer.

**Figure 2 biomolecules-14-00917-f002:**
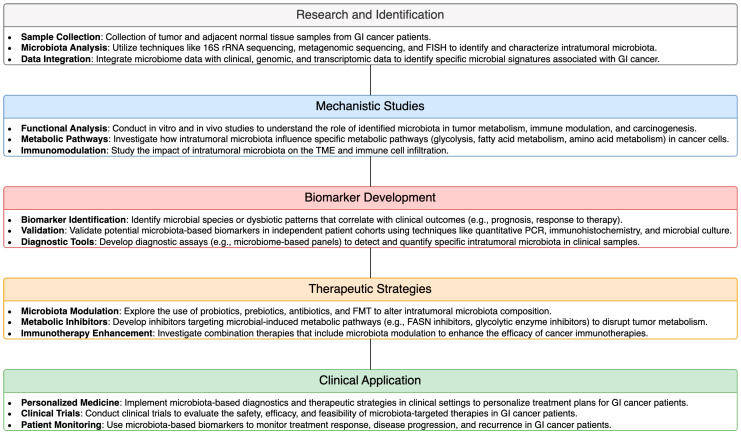
Schematic Pipeline of Therapeutic and Diagnostic Methodologies Utilizing Intratumoral Microbiota in GI Cancer.

**Table 1 biomolecules-14-00917-t001:** Summary of recent studies on intratumoral microbiota in pancreatic cancer.

Year of Study	Nature of Study	Participants	Aim and Main Findings	Refs.
2019	Experimental and observational study	PDAC patients; mice	The study aimed to investigate the role of the tumor microbiota and the immune system in influencing long-term survival (LTS) in patients with PDAC. The main findings reveal that patients with long-term survival had higher alpha-diversity in their tumor microbiome. An intra-tumoral microbiome signature (Pseudoxanthomonas-Streptomyces-Saccharopolyspora-Bacillus clausii) was identified as highly predictive of long-term survival. FMT experiments demonstrated that modulating the tumor microbiome could affect tumor growth and immune infiltration.	[57]
2018	Experimental study	PDAC patients; mice	The study aimed to investigate the role of the microbiome in PDAC and its potential as a therapeutic target. The study revealed that the cancerous pancreas harbors a significantly more abundant microbiome compared to the normal pancreas, with specific bacteria increased in tumorous pancreas compared to the gut. Ablation of the microbiome protects against preinvasive and invasive PDA, while transfer of bacteria from PDA-bearing hosts reverses tumor protection.	[58]
2022	Experimental study	PDAC cells; mice; PDAC patients	The study aimed to investigate the mechanisms by which type 2 immune cells traffic to the tumor microenvironment in PDAC and identify potential therapeutic targets. Intratumoral fungal mycobiome plays a crucial role in IL-33 secretion, and targeting this pathway shows therapeutic potential in reducing Th2 cells and innate lymphoid cells 2 (ILC2) infiltration and promoting tumor regression in PDAC.	[59]
2022	Experimental study	PCA patients; mice	The study aimed to investigate the role and mechanism of *P. gingivalis* in promoting PCA oncogenesis. The study explores a significant association between *P. gingivalis* and PCA, demonstrating its presence in both oral cavity and tumor tissues of PCA patients. Exposure to *P. gingivalis* accelerates tumor development in mouse models of PCA, fostering a neutrophil-dominated proinflammatory TME through elevated secretion of neutrophilic chemokines and neutrophil elastase (NE).	[60]
2024	Observational and experimental study	PDAC patients	The study aimed to elucidate the impact of intratumoral bacteria on the pathophysiology and prognosis of PDAC patients. The study identifies the presence of intratumoral bacteria, particularly anaerobic species such as Bacteroides, Lactobacillus, and Peptoniphilus, in human PDAC tissue, which is associated with suppressed anti-PDAC immunity and poorer prognosis.	[56]
2023	Experimental study with bioinformatics analysis	PDAC patients (data from the TCGA); mice	The study aimed to explore whether intervention with butyrate-producing probiotics can limit PDAC progression. The study demonstrates that intratumoral butyrate-producing microbiota is associated with favorable outcomes in PDAC. Intervention with *Clostridium butyricum* or its metabolite butyrate induces superoxidative stress and intracellular lipid accumulation, enhancing susceptibility to ferroptosis and inhibiting PDAC progression.	[61]
2023	Experimental and observational study	Mice	The study characterized the fecal and intratumoral microbiome of mouse models of PDAC and found significant differences compared to healthy controls. The fecal microbiome of KPC mice resembled that of human PDAC patients, and KPC tumors harbored more bacterial components compared to healthy pancreas tissue.	[62]
2020	Observational study using bioinformatics analysis	Pancreatic adenocarcinoma patients (data from the TCGA)	The study aimed to characterize the intra-pancreatic microbiome in pancreatic adenocarcinoma and explore its association with prognosis, smoking, and gender. The presence of specific bacterial species within pancreatic adenocarcinoma tumors was correlated with metastasis and immune suppression. Additionally, the study highlights the link between the increased prevalence and poorer prognosis of pancreatic adenocarcinoma in males and smokers with the presence of potentially cancer-promoting or immune-inhibiting microbes, emphasizing the importance of understanding and targeting the pro-TME for therapeutic interventions.	[63]
2021	Observational study	PDAC patients	This study investigated the role of clinical factors in bacterial colonization within PDAC. Findings reveal that biliary stent placement and neoadjuvant chemotherapy were associated with increased intratumor bacterial colonization, particularly from the Enterobacteriaceae family.	[64]
2023	Observational study	Patients with obstructive jaundice	This study aimed to define bile microbiome in patients with obstructive jaundice caused by PDAC compared to benign pancreaticobiliary diseases. Using 16S rRNA sequencing, distinct microbial signatures were identified, with PDAC patients exhibiting altered bile microbiome composition characterized by lower abundance of Escherichia and two unclassified genera, and increased abundance of Streptococcus.	[65]
2022	Case-control study	PDAC, pancreatic cyst(s), and normal pancreata patients	The case-control study aimed to investigate the duodenal fluid microbiome profiles in patients with PDAC, pancreatic cyst(s), and normal pancreata. Patients with PDAC exhibited diminished alpha diversity and enrichment of Bifidobacterium genera compared to control subjects and those with pancreatic cyst(s).	[66]
2023	Observational study	Intraductal papillary mucinous neoplasms (IPMNs) patients	This study investigated the association between specific microbiota and clinicopathologic characteristics of IPMNs of the pancreas. It found a higher relative abundance of Bacteroidetes and Fusobacteria in invasive IPMNs compared to noninvasive IPMNs, suggesting a potential role of intratumoral microbiota in the progression of IPMNs.	[67]
2022	Prospective pilot study	PDAC patients	This prospective pilot study demonstrates the feasibility of using endoscopic ultrasound-guided fine-needle biopsy to obtain adequate fresh tumor tissue for intratumoral microbial research in patients with PDAC. The intratumoral microbiome profiles generated from tissues obtained by EUS-FNB were comparable to those obtained by surgical biopsy, suggesting that EUS-FNB could serve as a valid and valuable tool for studying the intratumoral microbiome in both resectable and unresectable PDAC.	[68]
2022	Comparative study	PDAC patients	The study aimed to investigate the association between intratumor microbiome composition and patient survival, as well as the effect of specific microorganisms on tumor growth inhibition. The composition of the intratumor microbiome differs significantly between long-term and short-term survivors of PDAC in Chinese patients, with differential bacterial composition associated with metabolic pathways in the TME. Additionally, administration of Megasphaera enhances tumor growth inhibition when combined with anti-programmed cell death-1 (anti-PD-1) treatment in mice bearing 4T1 tumors.	[69]
2022	Experimental study	Normal pancreatic epithelial cells and PDAC cells	The study aimed to explore how *F. nucleatum* infection influences PDAC progression. *F. nucleatum* infection induces normal pancreatic epithelial cells and PDAC cells to secrete cytokines promoting tumor progression, including increased proliferation and invasive cell motility. Blocking GM-CSF signaling significantly limits the proliferative gains induced by infection.	[70]
2023	Observational study	PCA patients	The study aimed to identify prognostic microbes and their impact on PCA. The main findings reveal 26 prognostic genera and two microbiome-related subtypes (Mcluster A and B). Patients in Mcluster B had a worse prognosis and higher TNM stage and pathological grade. Patients in Mcluster A were more likely to benefit from CTLA-4 blockers and various chemotherapeutic agents. Additionally, a microbe-derived model for assessing outcomes showed good predictive performance, and the expression of LAMA3 and LIPH was associated with advanced stage and poor prognosis in PCA.	[71]
2023	Observational study	Pancreatic adenocarcinoma patients	The study aimed to understand how the presence of *P. gingivalis* in pancreatic adenocarcinoma affects the tumor microenvironment and immune response. The presence of *P. gingivalis* in pancreatic adenocarcinoma samples is strongly associated with specific immune cell gene expression patterns, as well as reduced chemical complementarity between *P. gingivalis* antigen and T-cell receptor sequences recovered from tumor samples.	[72]

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
