# Peer review of "Intratumoral Microbiota: Metabolic Influences and Biomarker Potential in Gastrointestinal Cancer"

_biomolecules, 2024, doi:10.3390/biom14080917_

Round 1

Reviewer 1 Report

Comments and Suggestions for Authors

The authors have done  outlined and described a detailed analysis of the tumor microenvironment and its contribution to the microbiota therein and vice versa. There are microbe to tumor and counter influences that provide a wide variety of interactions at the TME(Tumor Micro environment) level. The authors have provided a good understanding to the readers about the advantages of understanding the microbiota at the TME to generate better targeting of the tumor vis a vis the microbiota. Having said that there are a few comments for the authors. 

Major,

 1. The authors should add a table of the following

  Tumor-Microbe- positive or negative metabolic outcomes- therapeutic potential in that order in each column. 

2. The authors should also envision and provide a schematics of how these therapeutic/diagnostic methodologies present and future can be brought to  demonstrate how intratumoral microbiota can be used. From Research to Diagnosis a tentative pipeline would be useful.

Author Response

Answer: We thank the reviewer so much for the valuable feedback on our manuscript. We appreciate your insightful comments and have made the following revisions to address the concerns raised:

  1. We thank the reviewer so much for the valuable comments. We have revised and expanded Table 1 and Table 2 to include detailed information on the tumor-microbe research in GI cancers. Additionally, we would like to highlight that the content in section Intratumoral Microbiota and GI Cancer already encompasses a comprehensive summary of Tumor-Microbe interactions in GI cancers. This section provides a detailed analysis of various GI cancers and their associated microbial interactions, illustrating both beneficial and detrimental metabolic outcomes. We hope this enhances the overall understanding of the subject matter.
  2. We thank the reviewer so much for the valuable comments, we have added a new figure, titled "Figure 2. Schematic Pipeline of Therapeutic and Diagnostic Methodologies Utilizing Intratumoral Microbiota in GI Cancer." This schematic pipeline illustrates the envisioned transition from research to diagnosis and potential therapeutic applications, demonstrating how intratumoral microbiota can be effectively utilized.

The modified content is shown in red. We hope these modifications meet your expectations and contribute to the manuscript's clarity and comprehensiveness.

Reviewer 2 Report

Comments and Suggestions for Authors

Thank you to the authors for a well written and clinically relevant review. There are few minor issues to address. 

1. Check Lines 132-134 and 153 – 155 for completeness. To also check abbreviation FMT was defined previously before use in Line 135-136.

2. Subcategorize the information in Table 1 and Table 2 instead of presenting as just one paragraph describing each study. Create a number of columns like is done for reporting selected studies in systematic or scoping review. Among the items to be included should be year of study, nature of study, participants, aim and main findings. 

3. Challenges and Future Directions: Need to be consistent regarding using all capitals in headings compared to previous headings. 

4. Need to cite Figure 1 somewhere in the text before its appearance instead of the conclusion.

5. FMT is missing among the list of abbreviations.

Author Response

Answer: We appreciate the reviewer’s thorough and insightful comments on our manuscript so much. These suggestions significantly enhance the quality of our work. We have revised the manuscript accordingly:

  1. We have ensured the completeness of lines 132-134 and 153-155. The abbreviation FMT is now defined prior to its first usage in lines 134-135.
  2. We have revised Tables 1 and 2 by subcategorizing the information as suggested. The tables now include columns for the year of study, nature of study, participants, aim, and main findings to align with systematic or scoping review standards.
  3. We have ensured consistency in the use of all capitals in headings within the "Challenges and Future Directions" section to match the format of previous headings.
  4. We have added Figure 1 to the section “3. Role of Intratumoral Microbiota in Modulating GI Tumor Metabolism” as appropriate.
  5. FMT has been added to the list of abbreviations for completeness.

The modified content is shown in red. We hope these modifications meet your expectations and contribute to the manuscript's clarity and comprehensiveness.